# Precipitation Hardening at Elevated Temperatures above 400 °C and Subsequent Natural Age Hardening of Commercial Al–Si–Cu Alloy

**DOI:** 10.3390/ma14237155

**Published:** 2021-11-24

**Authors:** Ruoqi Li, Naoki Takata, Asuka Suzuki, Makoto Kobashi, Yuji Okada, Yuichi Furukawa

**Affiliations:** 1Department of Materials Process Engineering, Graduate School of Engineering, Nagoya University, Furo-cho, Chikusa-ku, Nagoya 464-8603, Japan; suzuki.asuka@material.nagoya-u.ac.jp (A.S.); kobashi.makoto@material.nagoya-u.ac.jp (M.K.); 2Toyota Motor Corporation, Toyota-cho, Toyota 471-8571, Japan; yuji_okada@mail.toyota.co.jp (Y.O.); yuichi_furukawa@mail.toyota.co.jp (Y.F.)

**Keywords:** Al–Si–Cu alloy, age-hardening, intermetallic phases, artificial aging, natural aging

## Abstract

The precipitation of intermetallic phases and the associated hardening by artificial aging treatments at elevated temperatures above 400 °C were systematically investigated in the commercially available AC2B alloy with a nominal composition of Al–6Si–3Cu (mass%). The natural age hardening of the artificially aged samples at various temperatures was also examined. A slight increase in hardness (approximately 5 HV) of the AC2B alloy was observed at an elevated temperature of 480 °C. The hardness change is attributed to the precipitation of metastable phases associated with the α-Al_15_(Fe, Mn)_3_Si_2_ phase containing a large amount of impurity elements (Fe and Mn). At a lower temperature of 400 °C, a slight artificial-age hardening appeared. Subsequently, the hardness decreased moderately. This phenomenon was attributed to the precipitation of stable θ-Al_2_Cu and Q-Al_4_Cu_2_Mg_8_Si_6_ phases and their coarsening after a long duration. The precipitation sequence was rationalized by thermodynamic calculations for the Al–Si–Cu–Fe–Mn–Mg system. The natural age-hardening behavior significantly varied depending on the prior artificial aging temperatures ranging from 400 °C to 500 °C. The natural age-hardening was found to strongly depend on the solute contents of Cu and Si in the Al matrix. This study provides fundamental insights into controlling the strength level of commercial Al–Si–Cu cast alloys with impurity elements using the cooling process after solution treatment at elevated temperatures above 400 °C.

## 1. Introduction

Aluminum (Al) alloys typically feature low densities, high specific strengths, high thermal conductivities, and superior corrosion resistances. Hence, these alloys are widely used in aerospace, automobile, and mechanical engineering and their related industries. In one of the representative applications of Al alloys in the automobile industry, Al cast alloys are used for making cylinder blocks and heads in diesel or gasoline engines [1,2,3,4]. Al–Si alloys are a general series of cast-type alloys [5] and are generally used as Al alloy components formed by the casting process. The high thermal conductivity and low density of Al–Si alloys make them favorable alternatives to cast iron in the fabrication of automotive engine components. The bolting portions of cylinder heads operating in the combustion cycle are often subjected to high stress. Thus, sufficient strength at temperatures ranging from ambient temperature to approximately 150 °C is required for Al cast alloys used in cylinder components of engines. The addition of Cu as an alloying element can improve the strength of Al–Si alloys, and Al–Si–Cu ternary alloys are generally used as conventional materials in cylinder heads fabricated via sand and gravity die casting processes [6,7,8,9,10]. One of the conventional Al–Si–Cu cast alloys is the AC2B alloy (denoted according to Japan Industry Standard: JIS), with a nominal composition of Al–6Si–3Cu (mass%) [5]. On being subjected to heat treatment, the AC2B alloy can achieve improved strength by precipitation hardening. T6 heat treatment (high-temperature solution treatment followed by water-quenching and subsequent artificial-aging treatment at relatively low temperatures) is often used for manufacturing cylinder head components [10,11,12,13,14,15,16,17]. In general, the solution treatment is carried out at high temperatures, above 500 °C (below the eutectic temperature in an Al-rich alloy composition) in order to dissolve soluble phases in the α-Al (FCC) matrix for the formation of a supersaturated solid solution, following water quenching. During artificial aging treatment at 100–200 °C, fine precipitates of intermetallic phases homogenously form in the matrix, contributing to an improvement in the mechanical performance of the AC2B alloy.

To realize the highly efficient production of cylinder head components, it is necessary to simplify the heat treatment process for Al–Si–Cu cast alloys. As an initial attempt to simplify the production process, the artificial aging treatment is planned to be removed. In order to achieve sufficient strength of the AC2B alloy without applying the artificial aging treatment, it is necessary to control the precipitation during cooling after the solution treatment (at elevated temperatures above 500 °C). The cooling-rate sensitivity and precipitation kinetics of Al alloys can be described using continuous cooling precipitation (CCP) diagrams. These CCP diagrams need to cover not only a wide range of cooling rates but also a high temperature range (from 200 °C to above 500 °C) [18]. The nominal composition of the AC2B alloy (Al–6%Si–2.5%Cu) is indicated on a 6% Si vertical section of the Al–Si–Cu ternary phase diagram, and its schematic CCP diagram is presented in Figure 1. In the commercial AC2B alloy (Figure 1a), the precipitation sequence of the Si and θ-Al_2_Cu phases was predicted during cooling after the solution treatment at an elevated temperature of approximately 540 °C (Figure 1b). In this regard, it is necessary to understand the precipitation kinetics and its associated age-hardening at a high temperature range (just below the solution treatment temperature, as shown in Figure 1b). However, there is little available research on the precipitation of Al–Si–Cu cast alloys at elevated temperatures above 400 °C. In the last decade, differential scanning calorimetry (DSC) was used to investigate the cooling rate sensitivity of exothermal reactions (precipitations) in Al alloys for preparing CCP diagrams [18,19,20,21]. However, previous studies have focused on age-hardenable wrought alloys (Al–Mg–Si alloys and Al–Zn–Mg alloys) to address the quench sensitivity of Al alloys after solution treatment and its effect on hardening after the artificial aging treatment.

In the industrial production of cylinder heads, maintaining the durability and quality of the mechanical performance of Al–Si–Cu cast alloys controlled by the cooling process is a key issue (a stable strength level is required for maintaining ambient temperature). It is assumed that subsequent hardening by maintaining the cooling-processed materials at ambient temperature (natural-age hardening [22,23,24,25,26]) might play a significant role in the stability of the room-temperature strength of the AC2B alloy. During the cooling process from the solution-treatment temperature, fine precipitates would nucleate and grow by consuming solute elements in the supersaturated solid solution of the α-Al matrix. The remaining solute atoms (in particular, Cu atoms) in the matrix may contribute to natural age hardening, which is likely due to the formation of atomic clusters or nanoscale precipitates at ambient temperature [22,23]. Accordingly, to control the strength of the AC2B alloy treated by cooling, it is essential to understand the relationship between various intermetallic phases precipitated at elevated temperatures [24,25,26] and the natural age-hardening behavior.

In the present study, the precipitation behavior and its associated hardening by artificial aging treatments in the high-temperature range above 400 °C are systematically investigated using a commercial AC2B alloy and compared with the constituent phases determined by thermodynamic calculations in Al-based multi-element systems. The natural age-hardening of the AC2B alloy samples, which were artificially aged at various temperatures, is also investigated. The chemical composition analysis of the α-Al matrix in the artificially aged samples is performed. These results are utilized to discuss the effect of precipitated phases at elevated temperatures on the natural age hardening in terms of solute elements in the α-Al matrix and to gain fundamental insights into controlling the precipitation of Al–Si–Cu cast alloys via the cooling process.

## 2. Experimental Procedures

In this study, AC2B alloy ingots were used as representatives of commercial Al–Si–Cu cast alloys applied to cylinder head components. The chemical composition of the AC2B alloy was measured using inductively coupled plasma atomic emission spectroscopy (ICP-AES). The nominal and measured compositions are presented in Table 1. The AC2B alloy contains not only Si and Cu as alloy elements, but also trace amounts of Mg, Fe, Mn, and Zn as impurity elements. The alloy ingots were cut into centimeter-sized samples for heat treatment. In order to determine the appropriate temperature of the solution treatment prior to the artificial aging treatment, the solution treatment was carried out in the range of 480–540 °C for different durations ranging from 3.6 × 10^3^ s (1 h) to 6.48 × 10^4^ s (18 h), followed by water quenching. The artificial aging treatment was subsequently conducted at various temperatures from 400 to 500 °C for different durations from 300 s to 3.6 × 10^4^ s (10 h). The isothermal aging experiments contributed to the prediction of the time-temperature-precipitation (TTP) diagram, which is closely related to the CCP diagram (as illustrated in Figure 1) [27]. The aging treatments were carried out immediately after the solution treatment to eliminate the effect of natural aging on precipitation during the artificial aging treatment.

To observe the microstructures, the sample surface was mechanically polished and then finished with colloidal silica (with mean particle size of 0.05 μm). The macroscopic constituent phases of the samples were identified using X-ray diffraction (XRD) analysis equipped with a Cu target tube (λ_Kα_ = 0.154 nm) (UltimaIV, Rigaku, Japan). The microstructures were characterized using a scanning electron microscope (SEM; JSM-6610A, JEOL Ltd., Tokyo, Japan) equipped with an energy dispersive spectroscopy (EDS) system. Thin foil samples for transmission electron microscopy (TEM) observations were prepared using an ion slicer (JEOL Ltd., Japan) at 6 V. The TEM observation and composition analyses were carried out using a JEM-2100HK system operating at 200 kV. The samples were studied using scanning transmission electron microscopy (STEM)-EDS analyses. The hardness (HV) values of the samples were measured using a Vickers indenter (FM-300e, FUTURE-TECH CORP, Kanagawa, Japan) at a constant load of 4.9 N and a dwell time of 15 s at ambient temperature. The hardness measurements were started within half an hour of the heat treatment to minimize the effect of natural age hardening on the measured hardness. More than five hardness measurement tests were performed for each sample.

## 3. Results

### 3.1. Solution Treatment

Figure 2 shows the change in the hardness of the AC2B alloy sample with holding times at different temperatures during the solution treatment. The as-fabricated sample exhibited a high hardness of approximately 100 HV. The hardness dropped to approximately 70 HV after 1 h (3.6 × 10^3^ s) and then remained almost constant at all solution temperatures (from 480 °C to 540 °C). The 480 °C solution-treated sample exhibited a slightly higher hardness than the samples solution-treated at higher temperatures (500 °C, 520 °C and 540 °C). The results indicate that solution treatment for longer than 1 h sufficiently reduces the hardness of the studied sample. The back-scattered electron (BSE) images (BEIs) showing the microstructures of the solution-treated samples at different temperatures for 1 h are presented in Figure 3. Two phases with different contrasts were observed in the α-Al matrix (dark contrast) of the solution-treated samples, whereas pores were observed around the elongated phase (with intermediate contrast) in the sample treated at 540 °C (Figure 3d). These pores indicate the local presence of a liquid phase at the location during solution treatment. The representative BSE image (BEI) and the corresponding EDS elemental maps of the sample treated at 520 °C are shown in Figure 4. The chemical composition analyses reveal the enrichment of Fe and Mn elements in the bright-contrast phase (Figure 4a–c) and Si enrichment in the intermediate contrast phase (Figure 4a,d), which revealed that the impurity Mn and Fe elements formed intermetallic phases. Figure 5 shows the representative XRD profiles of the 520 °C solution-treated sample (Figure 5a), together with the subsequently artificially aged samples (Figure 5b,c). High-intensity diffractions derived from the α-Al and α-Al_15_(Fe, Mn)_3_Si_2_ [28] phases were detected in the sample treated at 520 °C (Figure 5a). Relatively high-intensity peaks of the Si phase with a diamond structure were also found. These results demonstrate a microstructure composed of α-Al_15_(Fe, Mn)_3_Si_2_ (bright contrast) and Si (intermediate-contrast) phases in the α -Al matrix of the solution-treated samples. The observed morphology of the α-Al_15_(Fe, Mn)_3_Si_2_ phase corresponds well with the results of previous studies on commercial Al cast alloys [10,11,12,13,29,30]. In addition, no clear diffraction peaks derived from the β-Al_9_Fe_2_Si_2_ phase [31] were detected in the 520 °C solution-treated sample (Figure 5a) and subsequently aged samples (Figure 5b,c), indicating a slight formation of a β-Al_9_Fe_2_Si_2_ phase in the studied AC2B alloy samples. The β phase was often found in Al-Si alloy heat-treated at high temperatures above 500 °C [32]. Based on the aforementioned results, the solution treatment condition of 520 °C/18 h (6.48 × 10^4^ s) was fixed for the experiments on the artificial-age treatment.

### 3.2. Artificial Aging at Elevated Temperatures above 400 °C

Figure 6 shows changes in the hardness of the 520 °C solution-treated sample with artificial-aging time at various temperatures. Discernible age hardening was not observed at high temperatures ranging from 400 to 500 °C. Nevertheless, the hardness increased from 69 HV to a maximum of approximately 74 HV after the age treatment at 480 °C for 3.6 × 10^3^ s (1 h). Several experiments confirm the reproducibility of the small increase in hardness by aging at 480 °C. Further aging treatment decreased the hardness of the sample below 60 HV after 3.6 × 10^4^ s (10 h). Such hardening behavior was not observed in the sample aged at 450 °C and 500 °C. The hardness was almost constant (approximately 69 HV) up to 7.2 × 10^3^ s during aging at 500 °C and then reduced to approximately 67 HV. In the aging at 450 °C, the hardness appeared to decrease in line with the aging time. These results indicate that a small increase in hardness occurred by aging at an elevated temperature of 480 °C, whereas no obvious hardening was found at higher and lower temperatures (450 °C and 500 °C). Notably, in the aging at a lower temperature of 400 °C, the hardness also slightly increased to approximately 73 HV after 2.7 × 10^2^ s and then appeared to decrease moderately.

The BSE images (BEIs) showing the microstructures of the samples aged at various temperatures for 3.6 × 10^3^ s (1 h) and 3.6 × 10^4^ s (10 h) are shown in Figure 7. In the sample that was artificially-aged at 480 °C for 1 h (age-hardened sample, as shown in Figure 6), the α-Al_15_(Fe, Mn)_3_Si_2_ and the Si phases were observed in the α-Al matrix (Figure 7a) as well as the solution-treated sample (Figure 3). Although a fine granular α-Al_15_(Fe, Mn)_3_S_i2_ phase (bright contrast) appeared to form around the Si phase (intermediate contrast), such fine precipitates were scarcely observed within the α-Al matrix at the SEM resolution level. A similar microstructure was observed in the sample aged for a longer time of 10 h (Figure 7b), indicating a slight change in the microstructure after further aging at 480 °C. The trend was found in the samples aged at 450 °C (Figure 7c,d). By contrast, a number of fine precipitates (bright contrast) were distributed in the α-Al matrix of the samples aged at 400 °C for 1 h (Figure 7e). The EDS composition analyses revealed the enrichment of Cu element in the observed precipitates. The formation of Cu-rich intermetallic phases was in good agreement with the low-intensity diffractions derived from the Q-Al_4_Cu_2_Mg_8_Si_6_ phase [33] and the θ-Al_2_Cu phase [34] (Figure 5c). These precipitates might contribute to the slight hardening by aging at 400 °C (Figure 6). Coarsened precipitates were observed in the sample aged for 10 h (Figure 7f), suggesting that the coarsening of the precipitates might have reduced the hardness of the sample (Figure 6).

Figure 8 presents TEM bright-field images showing precipitates inside the α-Al matrix of the artificially aged sample at different temperatures. A number of fine precipitates with a mean size of approximately 200 nm were distributed in the sample aged at 480 °C for 1 h (Figure 8a), indicating that the observed small increase in hardness (Figure 6) is responsible for the fine precipitates in the α-Al matrix. After 10 h of aging at 480 °C (Figure 8b), the density of the fine precipitates appeared to be lower. This result suggests the dissolution of fine precipitates into the α-Al matrix at an elevated temperature of 480 °C. The reduced density of the precipitates corresponds well with the reduction in hardness after 10 h (Figure 6). Precipitates were observed in the samples aged at 500 °C (Figure 8c) and 450 °C (Figure 8d). However, the precipitation density was clearly lower than that of the sample aged at 480 °C. The results of the TEM observations rationalized the subsequent softening after the aging for 10 h (Figure 6). In the sample aged at a lower temperature of 400 °C, a different precipitation morphology was observed in the *α*-Al matrix. High-density, needle-shaped precipitates were observed, together with sparsely distributed granular-shaped precipitates with an average size of approximately 1 μm. The SEM-observed precipitates with a granular morphology correspond to the Cu-rich phase precipitated in the sample aged at 400 °C (Figure 7e). After 10 h of aging, the needle-shaped precipitates became coarsened, and their density decreased (Figure 8f). The granular-shaped precipitates also appeared coarsened. This result indicates that the coarsening of the two precipitates could have contributed to the reduction in hardness after 10 h of aging at 400 °C (Figure 6).

Figure 9 displays the representative results of the STEM-EDS analyses for the samples artificially aged at 480 °C and 400 °C for 1 h. As the high-angle annular dark field (HAADF) image contrast is sensitive to the atomic number [35], the bright contrast areas (fine precipitates) correspond to the enrichment of heavy elements in the sample aged at 480 °C (Figure 9a). The EDS composition analyses revealed the presence of the concentrated alloy elements of Fe, Mn, and Si in the precipitated phases (Figure 9b–d). The measured constituent elements suggest precipitates of the α-Al_15_(Fe, Mn)_3_Si_2_ phase [28]. In the sample aged at 400 °C, enriched Mg, Cu, and Si elements were detected in the needle-shaped precipitates (Figure 9e–h). This result implies the formation of a needle-shaped Q-Al_4_Cu_2_Mg_8_Si_6_ phase [33] at 400 °C. The precipitation morphology of the Q phase corresponds well with previous results for Al–Cu–Mg–Si alloys [8,9,10,36]. A high concentration of Cu was detected in the granular precipitates in the sample aged at 400 °C, indicating the coarsened precipitates of the θ-Al_2_Cu phase.

To identify the precipitates that contributed to age hardening (as shown in Figure 6) in the artificially aged sample at 480 °C for 1 h, electron diffraction analysis was performed. The results are presented in Figure 10. A TEM image of the precipitate in the α-Al matrix is shown in Figure 10a. The selected area electron diffraction (SAED) pattern (Figure 10b) indicates that the incident beam is parallel to the [011] direction in the α-Al matrix (B//[011]). The SAED pattern (Figure 10d) obtained from the location of the precipitate in the α-Al matrix shows a number of diffraction spots derived from the intermetallic phase. This diffraction pattern was in good agreement with the simulated diffraction pattern (Figure 10c) of the α-Al_15_(Fe, Mn)_3_Si_2_ phase [28]. Such diffraction patterns have been reported in Al-Si-Mg alloys (containing Fe and Mn elements) heated to 460 °C [37]. In the present study, the detailed crystal structure of the precipitates was not identified using limited zone axes presented in captured SAED patterns (Figure 10d). The differentiation among the stable α-Al_15_(Fe, Mn)_3_Si_2_ phase and its related metastable phases (α’(α_2_), α_v_, and α” phases [38,39]) remains unclear in the present characterization.

### 3.3. Natural Age-Hardening

Figure 11 shows the changes in the hardness of the samples that were artificially-aged at various temperatures for 3.6 × 10^3^ s (1 h) followed by holding at ambient temperature (natural aging). The increase in hardness corresponds to the natural age-hardening of the artificial-aged samples. The natural age-hardening behavior changed depending on the prior solution-treatment temperature or artificial-aging temperature (400–520 °C). The samples exhibited scattered hardness values (as shown in Figure 6) to some extent, whereas the hardness values were almost constant at about 70 HV until approximately 10^4^ s in natural-aging time (after the prior heat treatment). The hardness of the 520 °C solution-treated sample was found to significantly increase after natural aging for 104 s and eventually reached a value of approximately 108 HV. The hardness change was almost completed after 5 × 10^6^ s in all the samples. The maximum hardness obtained was lower than that of the samples that were subsequently artificially aged at various temperatures. The maximum hardness values of the 500 °C and 450 °C aged samples were approximately 103 HV, which was higher than that of the 480 °C aged sample (approximately 98 HV). It is clear that the 400 °C aged sample exhibited a much lower hardness, of approximately 75 HV. A similar result was found in the samples artificially aged at 400 °C or 480 °C for a longer time of 10 h. These results suggest that the natural-age hardening behavior would change depending on the precipitated phases formed by prior artificial-aging of the samples at different temperatures.

## 4. Discussion

### 4.1. Precipitation at Elevated Temperatures

The present study investigated precipitation in AC2B alloys and its associated hardness change at elevated temperatures above 400 °C. One of the important findings in this study is the occurrence of a small increase in hardness by aging at an elevated temperature of 480 °C (Figure 6). The increase in hardness could be associated with the precipitation of Fe- and Mn-rich phases in the α-Al matrix (Figure 8 and Figure 9). In the sample artificially aged at 400 °C, Cu-rich precipitates with a granular morphology (corresponding to θ-Al_2_Cu phase, as predicted in Figure 1) and fine needle-shaped precipitates with concentrated Mg, Si, and Cu elements were observed within the α-Al matrix (Figure 7 and Figure 8). These results clearly indicate the necessity of accounting for the effects of Fe, Mn, and Mg as impurities in the AC2B alloy (Table 1) on the precipitation at high temperatures above 400 °C. To understand the intermetallic phases precipitated at high temperatures, thermodynamic equilibrium calculations for the detailed composition of the AC2B alloy (Table 1) were performed using the CALPHAD approach (calculated by Pandat^TM^ Software (Version2019, CompuTherm LLC, Middleton, WI, USA)) [15,40,41], based on a commercial thermodynamic database (PanAluminum) [42]. The calculated results are summarized in Figure 12. The calculated phase diagrams indicate the constituent phases in the alloy composition (in equilibrium). In the Al–Si–Cu ternary system (major alloy elements), the solidus is located at approximately 540 °C and the two-phase region of the α-Al (FCC) and Si (diamond) phases are below the solidus temperature for the AC2B alloy composition (Figure 12a). The three-phase region of the α-Al + Si + θ-Al_2_Cu phase was below approximately 450 °C. In the Al–Si–Cu–Fe quaternary system (Figure 12b), the Fe-rich intermetallic β-AlFeSi phase (τ_6_-Al_9_Fe_2_Si_2_) [31] appears below approximately 560 °C and in equilibrium with the other solid phases (α-Al, Si, and θ-Al_2_Cu phases) at lower temperatures. The calculated results indicate that a trace amount (0.4% mass) of Fe (as an impurity) could enhance the formation of the β phase in Al–Si–Cu alloys, which is in good agreement with the results in previous reports on conventional Al–Si–Cu alloys [13,30]. The thermodynamic calculation for the Al–Si–Cu–Fe–Mg system (Figure 12c) assessed the formation of the Q-Al_4_Cu_2_Mg_8_Si_6_ phase [33] in the AC2B alloy composition containing a small amount of Mg (0.4% mass) at temperatures below approximately 470 °C, which corresponded well with the needle-shaped precipitates observed in the sample aged at 400 °C (Figure 8 and Figure 9). Furthermore, the calculation for the Al–Si–Cu–Fe–Mn–Mg system indicated that the addition of Mn would stabilize the α-Al_15_(Fe, Mn)_3_Si_2_ intermetallic phase [28] rather than the β-AlFeSi phase in the alloy composition, particularly at high temperatures. This calculation was consistent with the formation of the α-Al_15_(Fe, Mn)_3_Si_2_ phase observed in the solution-treated samples (Figure 3, Figure 4 and Figure 5). A comparison of the thermodynamic calculations (Figure 12) with the experimental results indicated the high reliability of the calculated constituent phases in the Al–Si–Cu–Fe–Mn–Mg multi-element system for the studied AC2B alloy composition.

Figure 13 presents the calculated variations in the constituent phase fractions (in equilibrium) at elevated temperatures using thermodynamic calculations for the Al–Si–Cu–Fe–Mn–Mg system. The calculated temperature dependence of constituent phase fractions is presented in Figure 13b. The calculated results indicate the presence of a liquid phase in the studied alloy composition at temperatures higher than approximately 540 °C. The presence of the liquid phase corresponds well with the results of the solution-treated samples (Figure 3). Below the solidus temperature, it was determined that the Si and the α-Al_15(_Fe, Mn)_3_Si_2_ phases with relatively high fractions were formed in the α-Al matrix. The β-AlFeSi phase exhibited fractions lower than 1% and appeared destabilized at lower temperatures. This trend is consistent with the α-Al_15_(Fe, Mn)_3_Si_2_ phase with high fractions observed in the solution-treated samples (Figure 3, Figure 4 and Figure 5). Notably, two Cu-rich intermetallic phases (θ-Al_2_Cu and Q-Al_4_Cu_2_Mg_8_Si_6_ phases) were in equilibrium with the α-Al phase below 470 °C, and their calculated fractions became higher at lower temperatures (Figure 13b). This is consistent with the number of granular Cu-rich phases and fine needle-shaped precipitates (containing Mg, Cu, and Si) observed in the 400 °C aged sample (Figure 9e–h). These results indicate that the slight age hardening at 400 °C (Figure 6) could be responsible for the precipitation of the θ-Al_2_Cu and Q-Al_4_Cu_2_Mg_8_Si_6_ phases. We found that the fractions of Si and α-Al_15_(Fe, Mn)_3_Si_2_ phases become higher at lower temperatures (in the studied temperature range of 400–520 °C), suggesting the precipitation of these phases at elevated temperatures above 400 °C. However, a number of fine precipitates with enriched Fe and Mn elements (no precipitates of Si phase) were observed in the 480 °C aged sample (Figure 8a and Figure 9a–d), whereas the number density was significantly lower in the sample aged for a longer duration (Figure 8b). These experimental results are indicative of the dissolution of precipitates into the α-Al matrix, contributing to the observed softening after 10 h of aging (Figure 6). The difference between the thermodynamically calculated and experimental results was likely due to the formation of metastable phases at the early stage of precipitation at 480 °C. The electron diffractions derived from the precipitates corresponded well with the lattice planes of the α-Al_15_(Fe, Mn)_3_Si_2_ phase (Figure 10), suggesting the precipitation of metastable phases associated with the α phase. In fact, various types of metastable intermetallic phases (α’(α_2_), α_v_, and α” phases [38,39]) have been reported for Al–Si–Fe-based alloys. Although the observed metastable phase could not be crystallographically identified in the present study, the fast kinetics of the precipitation of the metastable α-associated phases (consuming solute Fe, Mn, and Si elements in the α-Al matrix) can be considered to contribute to the small increase in hardness observed at 480 °C (Figure 6). The metastable precipitates would dissolve into the matrix after further aging, resulting in softening by over aging at 480 °C (Figure 6).

The aforementioned calculated and experimental results can provide the precipitation sequence in the studied AC2B alloy at elevated temperatures above 400 °C. The dissolution of the precipitates after holding at 480 °C might lead to the formation of a stable α-Al_15_(Fe, Mn)_3_Si_2_ phase. The precipitation of the stable α phase might follow the precipitation of metastable phase. At temperatures below 450 °C, the precipitation of Cu-rich θ and Q phases may begin earlier by 1 h than at 400 °C. In general, much finer precipitates of θ and Q phases have often been observed in Al–Si–Cu-based alloys aged at lower temperatures [8,9,10]. These results provide the possibility of controlling the strength of the alloy through the precipitation of the metastable phases during cooling after the solution treatment. Furthermore, it suggests the significance of controlling Mn and Fe elements (for stabilizing the α-Al_15_(Fe, Mn)_3_Si_2_ phase) as impurities in the AC2B alloy to maintain sufficient strength without applying artificial age treatment. However, detailed the TTP diagram (and CCT diagram) still remained unclear in the present study. It could be necessary to perform the thermal analyses using differential scanning calorimetry (DSC) for identifying the precipitation sequence in detail.

### 4.2. Natural-Age Hardening at Ambient Temperature

The present study demonstrated that natural age-hardening behavior significantly changed depending on the prior artificial aging temperatures ranging from 400 °C to 500 °C (Figure 11). The 400 °C artificially aged sample exhibited a slight natural-age hardening; a number of Cu-rich intermetallic phases (θ-Al_2_Cu and Q-Al_4_Cu_2_Mg_8_Si_6_ phases) were finely precipitated at 400 °C (Figure 9e–h), resulting in a reduced solute Cu element in the α-Al matrix. It is generally understood that, in solution-treated Al alloys, solute elements (and/or vacancies) in the α-Al matrix could be closely associated with the formation of atomic clusters at ambient temperature, contributing to natural age hardening [22,23,24,25,26]. The present results suggest that natural age hardening is controlled by the precipitation of intermetallic phases occurring at elevated temperatures (prior to artificial aging). Notably, the maximum hardness of the 480 °C artificially aged sample was to some extent lower than those of samples artificially aged at different temperatures of 450 °C and 500 °C (Figure 11). This indicates that the reduced solute elements (Fe, Mn, and Si) in the α-Al matrix (consumed by precipitation of metastable α-associated phases) would lead to a reduced maximum hardness. It can be assumed that not only Cu but also other elements (Si, Fe and Mn) might contribute to the formation of solute clusters in the AC2B alloy.

To investigate the effect of solute elements in the α-Al matrix on the natural-age hardening, STEM-EDS composition analyses for the α-Al matrix in the artificially aged samples (1 h and 10 h) were performed. Figure 14 presents the experimentally measured content of various elements in the α-Al matrix plotted as a function of the artificial-age temperature (“520 °C”), which corresponds to the solution-treatment temperature. In the present study, more than ten measurements were captured at different locations of the α-Al phase in each sample, providing the average elemental content values plotted in Figure 14. The calculated solubility limits for each element in the α-Al phase are shown by the broken lines for comparison with the experimental data. The measured concentrations of Cu and Si in the α-Al matrix changed depending on the artificial aging (or solution treatment) temperature (Figure 14). The trend of variation in the experimental values appeared similar to the calculated temperature dependence of the solubility limit in the α-Al matrix. The solubility limit values of Fe and Mn were calculated to be close to zero, which corresponds to a trace amount (below 0.2%) of detectable Fe and Mn elements in the α-Al matrix of all samples. Mg was not detected in any of the samples. These similar trends between the calculated and experimental values suggest that the observed microstructures almost reached the equilibrium state after artificial aging for more than 1 h. In the 400 °C artificially aged samples, there was a slight change in the measured Si and Cu contents between 1 h and 10 h of aging time (Figure 14). This result clearly indicates the coarsening of the precipitates of the stable θ and Q phases (driven by interfacial energy rather than chemical driving force [43]) at 400 °C, which is in good agreement with the SEM and TEM observations (Figure 7 and Figure 8). In the 480 °C artificially aged samples, the Cu solute content slightly changed depending on the aging time (1 h and 10 h) as well (Figure 14). However, the Si solute content in the sample aged for 1 h was somewhat lower (by approximately 0.5%) than that of the sample aged for 10 h. It is noteworthy that the measured Fe solute content was less than 0.2% (presumably below reliable element contents detected by EDS analyses) in most of samples, whereas a higher Fe content than 0.3 % was detected in the samples aged at 480 °C for 10 h. The increased solute content of Si and Fe after further aging can support the assumption of dissolution of metastable α -associated phases occurring during long-term aging at 480 °C (as discussed in Section 4.1). Thus, the STEM-EDS composition analyses revealed that the solute contents of alloy elements (including impurities) varied depending on the type of intermetallic phases precipitated in the α-Al matrix and its associated solubility limit.

Assuming that reliable values of Cu and Si solute contents in the α-Al matrix are measured by STEM-EDS analyses, the maximum hardness values of the studied samples (caused by natural age hardening, as presented in Figure 11) were plotted as functions of the measured solute contents. The results are shown in Figure 15. The summarized results clearly show that the hardness enhanced by natural age hardening is strongly dependent on the solute contents of Cu and Si. This indicates the possibility of controlling natural-age hardening by changing the types of precipitates with different constituent elements (for controlling the solute contents of alloy elements in the α-Al matrix) via a heat-treatment process (including the cooling process) at elevated temperatures above 400 °C. However, the effect of the detailed compositions of the other impurity elements (Fe, Mn, and Mg) in the α-Al matrix remained unclear in the studied AC2B alloy. It was assumed that these impurity elements (in particular, Mg [24,25,26]) might contribute to the formation of solute clusters contributing to natural age hardening. It was difficult to detect sufficient concentrations for the quantification of solute impurity elements in the α-Al matrix. To fully understand the high-temperature precipitation effect on the natural age hardening of commercial Al–Si–Cu alloys, it is necessary to fundamentally investigate the precipitation of intermetallic phases at elevated temperatures and the subsequent natural-age hardening using Al–Si–Cu-based alloys with controlled Fe, Mn, and Mg contents.

## 5. Conclusions

In this study, the precipitation of intermetallic phases and the associated hardness change by artificial aging treatments in the high-temperature range above 400 °C in the AC2B alloy (a model of commercial Al–Si–Cu cast alloys applied to cylinder head components) were systematically investigated. The natural age hardening of the AC2B alloy samples artificially aged at various temperatures was investigated. The following key results were obtained.

(1)A small increase in the hardness of the AC2B alloy was observed at an elevated temperature of 480 °C. No obvious hardening was observed at higher and lower aging temperatures (450 and 500 °C). The small increase in hardness was attributed to the precipitation of metastable phases associated with the α-Al_15_(Fe, Mn)_3_Si_2_ phase containing a large amount of impurity elements (Fe and Mn). At a lower temperature of 400 °C, slight age hardening appeared, followed by a moderate decrease in hardness. This phenomenon was attributed to the precipitation of stable θ-Al_2_Cu and Q-Al_4_Cu_2_Mg_8_Si_6_ phases and their coarsening after a longer duration. The precipitation sequence was rationalized by thermodynamic calculations for the Al–Si–Cu–Fe–Mn–Mg system. These results provide the possibility of maintaining sufficient strength of commercial Al–Si–Cu (+Mg) cast alloys by controlling Mn and Fe impurities.(2)The natural-age hardening behavior significantly varies depending on the prior artificial aging temperatures ranging from 400 °C to 500 °C. The maximum hardness changed from 75 to 105 HV. The hardness enhanced by natural age hardening is strongly dependent on the solute contents of Cu and Si in the α-Al matrix. These results demonstrate an approach to the control of natural-age hardening by changing the types of precipitates with different constituent elements (for controlling the solute alloy contents) via a cooling process at elevated temperatures above 400 °C.

## Figures and Tables

**Figure 1 materials-14-07155-f001:**
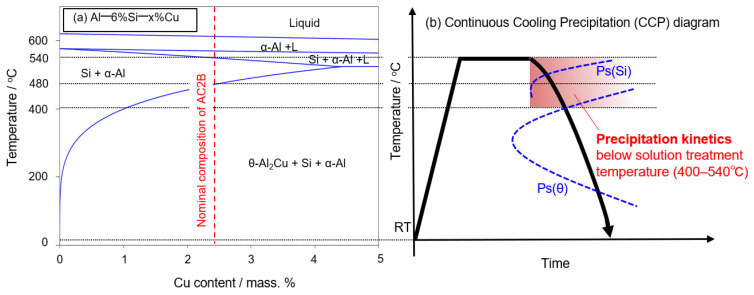
(**a**) Nominal composition of AC2B alloy (Al–6%Si–2.5%Cu) indicated on a 6% Si vertical section of the Al–Si–Cu ternary phase diagram and (**b**) schematic of the continuous cooling precipitation (CCP) diagram of the AC2B alloy during cooling (after the solution treatment at 540 °C).

**Figure 2 materials-14-07155-f002:**
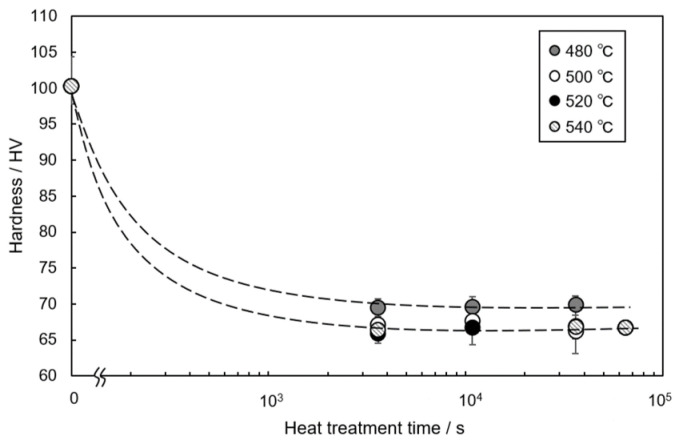
Change in hardness of the AC2B alloy sample with holding time at different temperatures of the solution treatment.

**Figure 3 materials-14-07155-f003:**
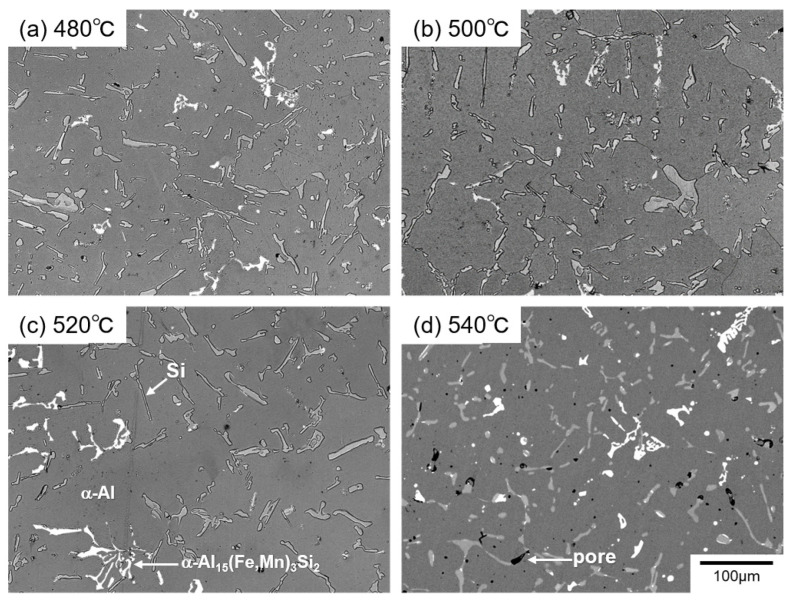
BSE images (or BEIs) showing microstructures of the AC2B alloy sample solution-treated at (**a**) 480 °C, (**b**) 500 °C, (**c**) 520 °C, and (**d**) 540 °C for 3.6 × 10^3^ s (1 h).

**Figure 4 materials-14-07155-f004:**
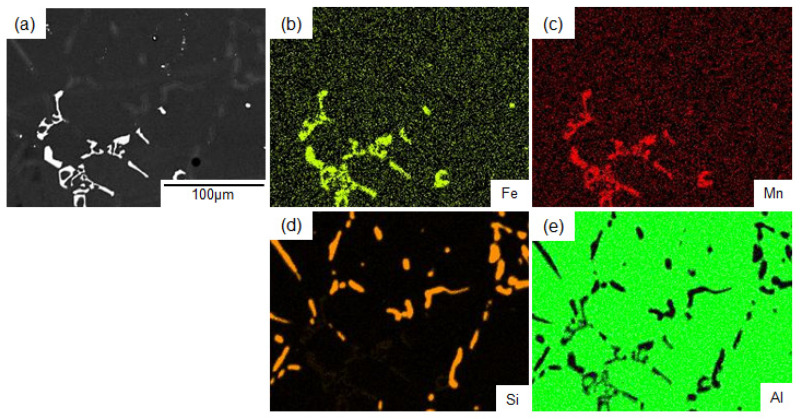
BSE image (BEI) of the AC2B alloy sample (**a**) solution-treated at 520 °C for 6.48 × 10^4^ s (18 h) and corresponding EDS element maps of (**b**) Fe, (**c**) Mn, (**d**) Si and (**e**) Al elements.

**Figure 5 materials-14-07155-f005:**
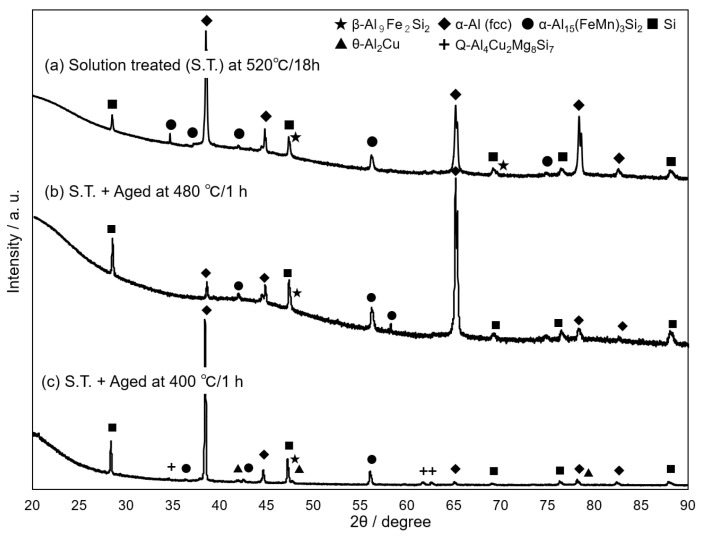
Representative XRD profiles of (**a**) 520 °C solution-treated AC2B alloy sample and (**b**,**c**) the samples subsequently aged at (**b**) 480 °C and (**c**) 400 °C.

**Figure 6 materials-14-07155-f006:**
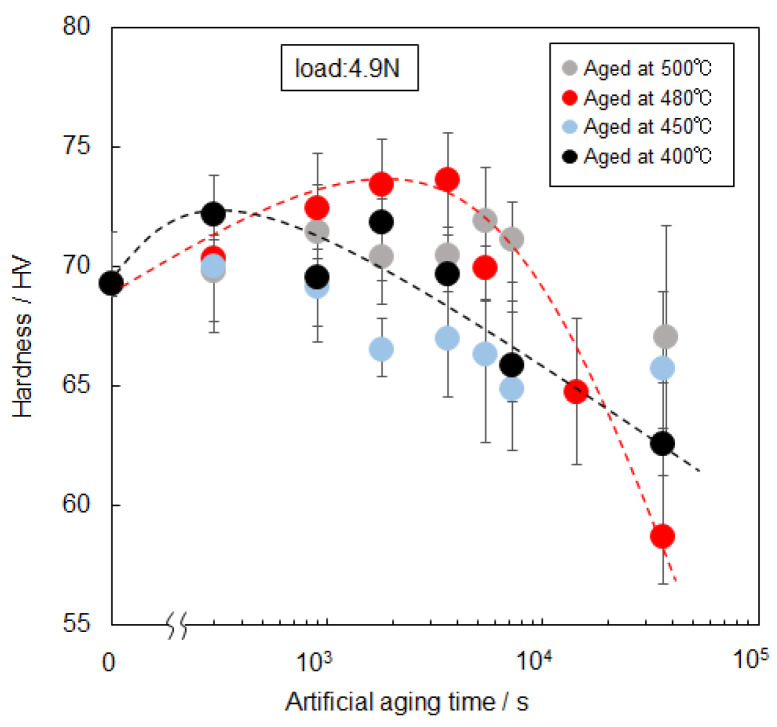
Changes in the hardness of the 520 °C solution-treated sample with artificial-aging time at elevated temperatures ranging from 400 °C to 500 °C.

**Figure 7 materials-14-07155-f007:**
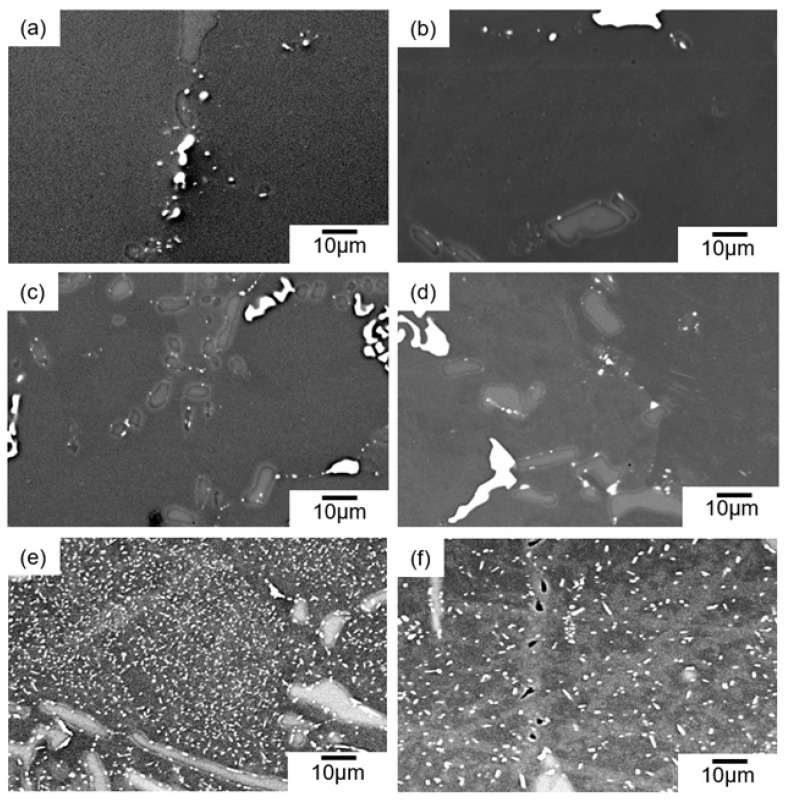
BSE images (BEIs) showing the microstructures of the samples aged at 480 °C for (**a**) 1 h and (**b**) 10 h, at 450 °C for (**c**) 1 h and (**d**) 10 h, and at 400 °C for (**e**) 1 h and (**f**) 10 h.

**Figure 8 materials-14-07155-f008:**
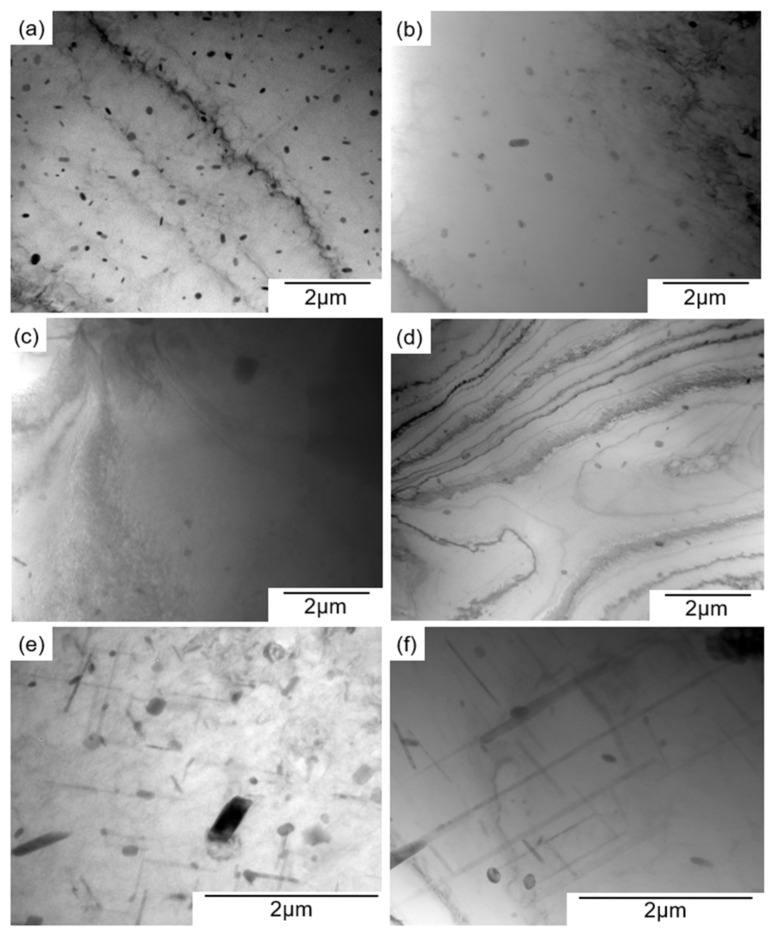
TEM bright-field images of samples aged at (**a**) 480 °C for 1 h, (**b**) 480 °C for 10 h, (**c**) 450 °C for 1 h, (**d**) 500 °C for 1 h, (**e**) 400 °C for 1 h, and (**f**) 400 °C for 10 h.

**Figure 9 materials-14-07155-f009:**
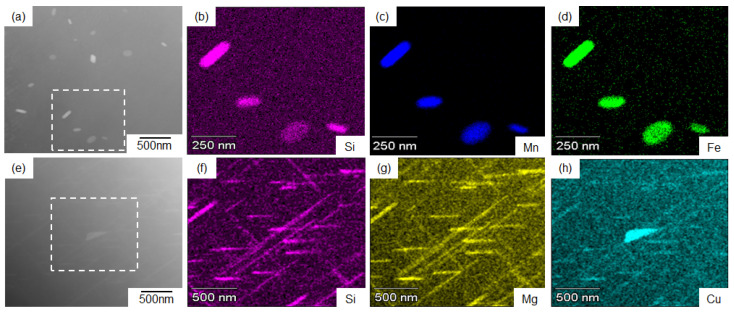
(**a**,**e**) High-angle annular dark field (HAADF)-scanning TEM (STEM) image and (**b**–**d**,**f**–**h**) the corresponding EDS element maps of AC2B alloy samples aged at (**a**–**d**) 480 °C for 1 h and (**e**–**h**) 400 °C for 1 h.

**Figure 10 materials-14-07155-f010:**
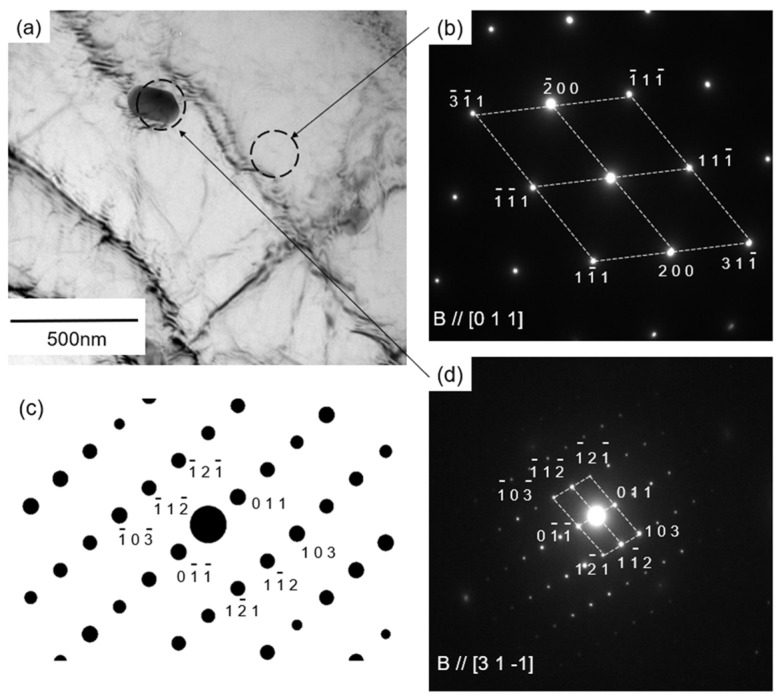
(**a**) TEM bright field images showing precipitates in the α-Al matrix. (**b**,**d**) The corresponding selected area diffraction patterns obtained from the samples aged at 480 °C for 1h, together with (**c**) the simulated diffraction pattern of α-Al_15_ (Fe, Mn)_3_Si_2_ phase.

**Figure 11 materials-14-07155-f011:**
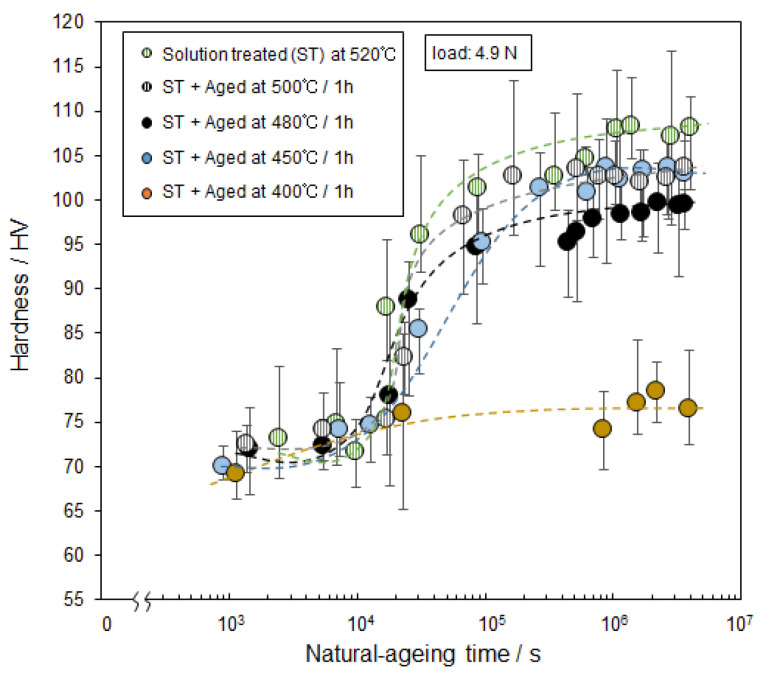
Changes in the hardness of the solution-treated sample and the samples artificially aged at different temperatures with holding time at room temperature (natural aging time).

**Figure 12 materials-14-07155-f012:**
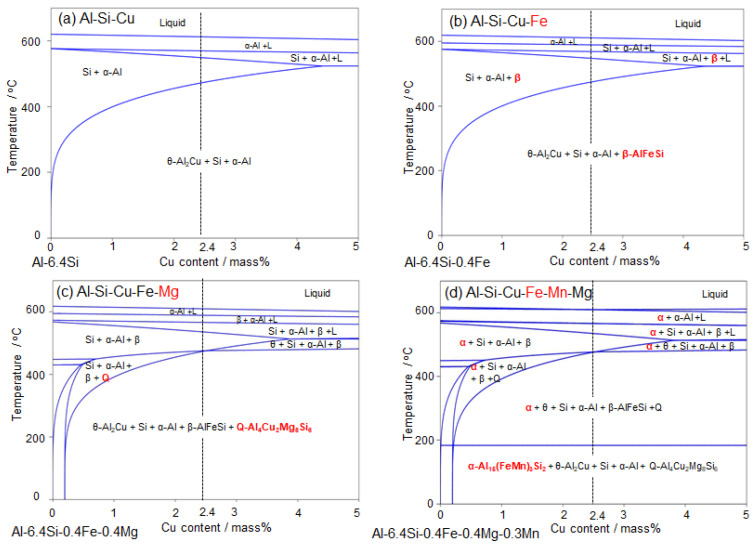
Calculated phase diagrams for the studied composition of AC2B alloy in (**a**) Al–Si–Cu system, (**b**) Al–Si–Cu–Fe system, (**c**) Al–Si–Cu–Fe–Mg system, and (**d**) Al–Si–Cu–Fe–Mn–Mg system.

**Figure 13 materials-14-07155-f013:**
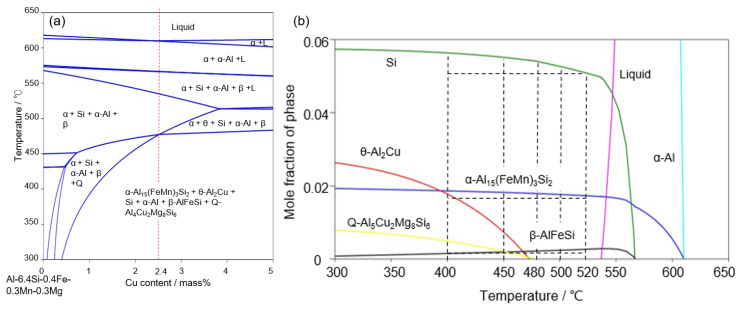
(**a**) Studied AC2B alloy composition indicated in a vertical section of the calculated Al–Si–Cu–Fe–Mn–Mg phase diagram and (**b**) changes in calculated phase fractions (in equilibrium) in the studied composition of AC2B alloy as a function of temperature.

**Figure 14 materials-14-07155-f014:**
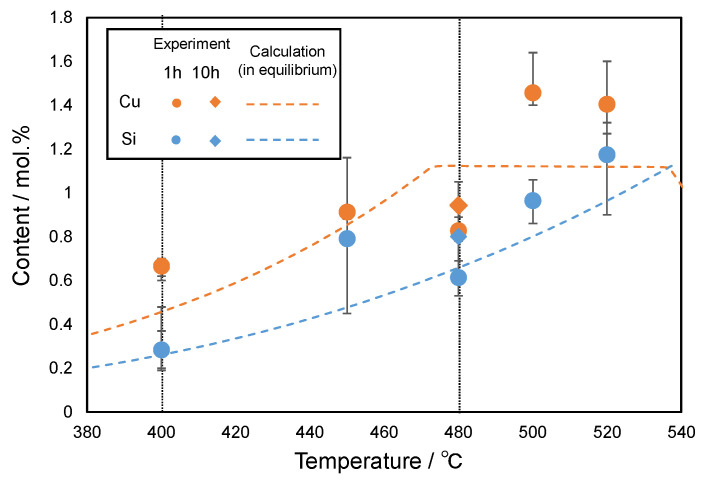
STEM-EDS measurement of the content of major alloy elements of Cu and Si in the α-Al matrix plotted as a function of artificial-aging temperature, together with calculated solubility limits for each element in the α-Al phase (in equilibrium).

**Figure 15 materials-14-07155-f015:**
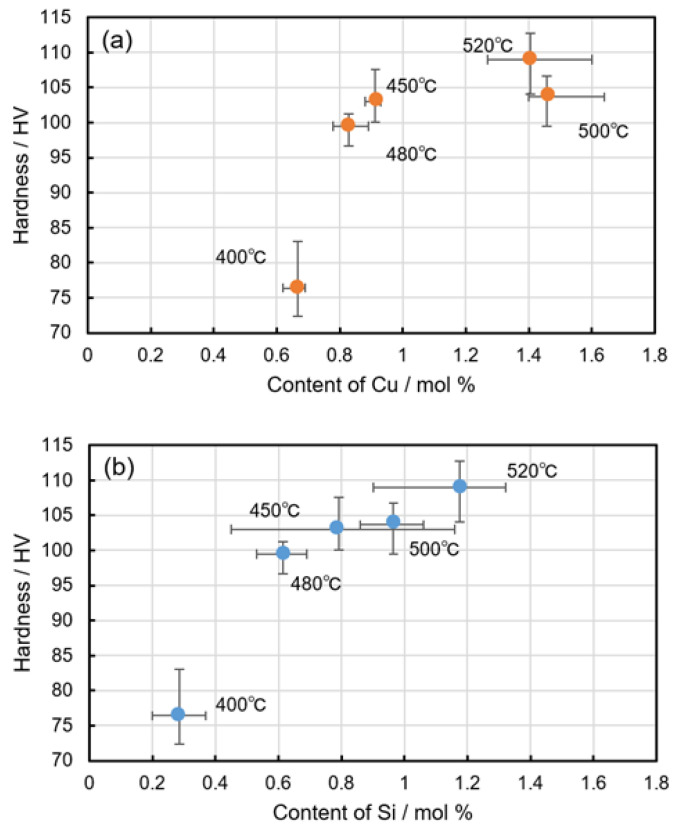
Maximum hardness values of the artificially aged (caused by natural-age hardening) AC2B alloy samples plotted as functions of measured solute content in the α-Al matrix.

**Table 1 materials-14-07155-t001:** Nominal and measured compositions of the studied AC2B alloy (mass%).

	Cu	Si	Mg	Zn	Fe	Mn	Ni	Ti	Pb	Sn	Cr
Nominal	2.0–4.0	5.0–7.0	≤0.5	≤1.0	≤1.0	≤0.5	≤0.35	≤0.2	≤0.2	≤0.1	≤0.15
ICP analyzed	2.40	6.35	0.29	0.28	0.42	0.30	0.03	0.05	0.01	0.01	0.03

## Data Availability

The data presented in this study are available in this article.

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
