# Peer review of "Precipitation Hardening at Elevated Temperatures above 400 °C and Subsequent Natural Age Hardening of Commercial Al–Si–Cu Alloy"

_materials, 2021, doi:10.3390/ma14237155_

Round 1
Reviewer 1 Report
The high thermal conductivity and low 35 density of Al–Si alloys make them favorable alternatives for cast iron in the fabrication of 36 automotive engine components. The paper deals with "Precipitation hardening at elevated temperatures above 400°C 2 and subsequent natural age hardening of commercial Al–Si–Cu 3 alloy", some results have been obtained. However, the current version still needs to be improved as follows,
(1) The tensile properties of the alloys in some typical conditions should be measured in this paper.
(2) The SEM tensile fracture should be also characterized in the paper.
(3) The relationship between microstructure and mechanical properties should be deeply discussed in the manuscript.
Author Response
We really appreciate the reviewer for giving useful comments to our manuscript. This paper focused on the fundamentals of precipitation of intermetallic phase at elevated temperatures above 400°C. The present study aims to provide fundamental insights into controlling the strength level of commercial Al–Si–Cu cast alloys with impurity elements using the cooling process after solution treatment at elevated temperatures. As the reviewer has pointed out, the mechanical properties of Al alloys controlled by cooling process is required in the next stage. It must await our future works to address the relation between precipitation morphology and mechanical properties (strength and ductility). We would like to report the result in our following paper.
Reviewer 2 Report
The study “Precipitation hardening at elevated temperatures above 400°C and subsequent natural age hardening of commercial Al–Si–Cu alloy” was carefully analyzed by reviewer. The study corresponds well to the journal scope, has advanced design, and provides scientific novelty. The reviewer does not mind if the current research is published in present state.
The minor notes are provided as follows:
There is an arrow in the Fig.3 (a) without a subscription. Should be corrected
The font in figure 5 should be increased.
From the reviewer viewpoint, the correct abbreviation for the back-scattered electron images is BSE images (not BEIs)
Sincerely yours,
Author Response
We really apologized for the carelessness on the figures, and those mistakes have been corrected. Regarding the problem of abbreviations, we believe that neither ‘back-scattered electron images (BEIs)’ nor ‘back-scattered electron (BSE) images’ is wrong, because both abbreviations are used in the papers. Revised words and sentences are highlighted (yellow) in the updated manuscript.
Reviewer 3 Report
The article is interesting and well organised on modeling the strength level of commercial Al–Si–Cu cast alloys with impurity elements using the cooling process after solution treatment at elevated temperatures above 400°C.
The article has practical importance for industry.
Please enlarge the images.
What CALPHAD approach was used? what software, please detail.
Make sure all infrastrucure is presented with details of model and mode of operation, e.g. what Vickers indenter ?
More discussions could be presented, e.g.:
Line 147-159:
" The chemical composition analyses reveal the enrichment of Fe and Mn elements in the bright-contrast phase (Fig. 4(a–c)) and Si enrichment in the intermediate contrast phase (Fig. 4(a, d))" why is that? from there this separation?
In general the article is a good technical approach.
Author Response
Thank you for your helpful suggestions. All images have been enlarged according to the reviewer’s comments. The details of all the model of infrastructure and the thermodynamic calculation software (PandatTM Software, CompuTherm LLC, Middleton) have been added in the revised manuscript. The separation of Si and Fe/Mn elements (Fig. 4) indicated the formation of Si (diamond) phase and alpha-Al15(Fe, Mn)3Si2 intermetallic phase in the sample solution-treated. Its related description has been added at lines 147-159 in the updated manuscript. All revised words and sentences are highlighted (yellow) in the updated manuscript. Please check them.